# Trophic Interactions of *Ceutorhynchinae* spp. (Coleoptera: Curculionidae) with Their Host Plants (Brassicaceae) and Their Parasitoids in the Agroecosystem of Quebec, Canada

**DOI:** 10.3390/insects14070607

**Published:** 2023-07-05

**Authors:** Claudine Desroches, Joseph Moisan-De Serres, Émilien Rodrigue, Geneviève Labrie, Éric Lucas

**Affiliations:** 1Laboratoire de Lutte Biologique, Département des Sciences Biologiques, Université du Québec à Montréal, Montréal, QC H2X 1Y4, Canada; noxdesroches@gmail.com; 2Laboratoire D’expertise et de Diagnostic en Phytoprotection (LEDP), Ministère de l’Agriculture, des Pêcheries et de l’Alimentation (MAPAQ), Québec, QC G1P 3W8, Canada; joseph.moisan-deSerres@mapaq.gouv.qc.ca (J.M.-D.S.); emilien.rodrigue@mapaq.gouv.qc.ca (É.R.); 3Centre de Recherche Agroalimentaire de Mirabel, Mirabel, QC J7N 2X8, Canada; glabrie@cram-mirabel.com

**Keywords:** natural enemy, plant-insect interactions, weevil, trophic web, Pteromalidae

## Abstract

**Simple Summary:**

The genus *Ceutorhynchus* Germar (Coleoptera: Curculionidae) is composed of canola pests, natural enemies of Brassicaceae, and other species associated with non-crop and non-weed plants. This study aimed to establish trophic associations of *Ceutorhynchus* with their host plants and with their parasitoids in the agricultural landscape, in order to assess the actual beneficial or noxious ecological roles of the insects. Trophic associations were established by identifying *Ceutorhynchus* species and their parasitoids emerging from collected Brassicaceae plants in areas adjacent to canola fields and other crops in 2019 and 2020. Two functional groups were distinguished: natural enemies of weeds and agricultural pests, and new trophic associations were described.

**Abstract:**

The genus *Ceutorhynchus* Germar (Coleoptera: Curculionidae) is composed of canola pests, natural enemies of Brassicaceae, and other species associated with non-crop and non-weed plants. This study aimed to establish trophic associations of *Ceutorhynchus* with their host plants and with their parasitoids in the agricultural landscape, in order to assess the actual beneficial or noxious ecological roles of the insects. Trophic associations were established by identifying *Ceutorhynchus* species and their parasitoids emerging from collected Brassicaceae plants in areas adjacent to canola fields and other crops in 2019 and 2020. Five *Ceutorhynchus* species were collected and identified as hosts of parasitoids in the families Pteromalidae and Eulophidae. Two functional groups were characterized: natural enemies of weeds and agricultural pests. The exotic wormseed wallflower, *Erysimum cheiranthoides* was identified as a new host plant of the invasive canola pest *Ceutorhynchus obstrictus* (Marsham), and the native tower rockcress, Arabis glabra, as a new host plant of the native *Ceutorhynchus neglectus* Blatchley. Association between the exotic *Ceutorhynchus typhae* (Herbst) and a parasitoid of the genus Elachertodomyia is reported for the first time. Finally, *Ceutorhynchus neglectus* and *C. typhae* hosted the exotic parasitoid *Trichomalus perfectus*, an important natural enemy of *C. obstrictus*.

## 1. Introduction

*Ceutorhynchus* Germar (Coleoptera: Curculionidae) are phytophagous weevils mostly associated with Brassicaceae [1]. They constitute the largest genus of Ceutorhynchinae representing about a quarter of all known species of this sub-family [2]. In the Holarctic region, 360 species of *Ceutorhynchus* have been reported [3]. The majority of *Ceutorhynchus* species lay eggs in pods (siliques), stems, or root crowns of their host plants [4,5]. When ready to pupate, larvae open a hole in the silique, stems or root crown and exit the plant to pupate in the soil [4,5].

There are few studies on the trophic relationships and ecology of the *Ceutorhynchus* assemblage in Canada. From a fundamental point of view, it is important to study and follow these abundant species, their host(s), and their natural enemies to understand how these communities are structured [6]. Furthermore, such a rich group may constitute a sentinel assemblage for recording large factors such as global warming impact, or biological invasion [7,8].

From an applied perspective, in Quebec’s agroecosystem, the *Ceutorhynchus* assemblage included canola pests, natural enemies of Brassicaceae weeds, and other species associated with non-crop and non-weed plants [9]. Then, studying herbivore-plant interactions of this taxon can provide useful insight into a biocontrol perspective (control of weeds by natural enemies) [10]. In order to assess the actual beneficial or noxious ecological roles of the assemblage, the study should include trophic interactions between *Ceutorhynchus* and higher trophic levels composed mainly of parasitoid community [11,12,13].

In Eastern Canada, the parasitoids community associated with *Ceutorhynchus* has only been studied by Mason et al. [5,14]. Two exotic parasitoids species, Mesopolobus gemellus and Trichomalus perfectus, are associated with native species of *Ceutorhynchus* in Ontario, and tracking these exotic parasitoids in Eastern Canada to monitor their spread and impact on non-target *Ceutorhynchus* has been recommended [5]. Those parasitoids are solitary ectoparasitoids that attack the concealed larvae of *Ceutorhynchus* [5]. The female parasitoid lays her egg inside the silique, stem or root crown with her ovipositor and injects a venom that paralyzes the host [5] The parasitoid completes its pupation inside the silique, stem, or root crown and will emerge by cutting a clean hole in the plant [5].

The main objective of the present study was to establish trophic associations of *Ceutorhynchus* with their host plants and with their parasitoids in the agricultural landscape of Eastern Canada. More specifically, our objectives were to: 1. Identify the *Ceutorhynchus* host plant and level of infestation, 2. Characterize the composition of their parasitoid guilds and determine levels of parasitism of the different *Ceutorhynchus* species, and 3. Describe the trophic web, linking *Ceutorhynchus* species to their parasitoids and host plants.

## 2. Materials and Methods

### 2.1. Sampling

Brassicaceae plants were collected during the summers of 2019 and 2020 in six regions of Eastern Canada (Quebec): Montérégie (M), Centre-du-Québec (CQ) and Chaudière-Appalaches (CH), Bas-Saint-Laurent (BSL), Saguenay Lac-Saint-Jean (SLSJ) and the Capitale-Nationale (CN). Those regions were visited 2 to 5 times per year. Regions closer to Montreal, where we were based, were visited 5 times (M, CQ, CH). Plants were collected from the middle of June to the end of August in 2019. Since no parasitoids emerged from plants collected in June 2019, in 2020, plants were only collected from the beginning of July to the end of August. Brassicaceae plants were collected in borders near canola fields, either beside canola fields or other crops.

Brassicaceae plants were collected at 8 different sites per region. Each region had 4 sites bordering canola fields and 4 sites bordering other crops. For every site and at every visit, the first five plants observed of any Brassicaceae species observed were collected. Not all species of Brassicaceae were collected in every site and every region. In some sites (around 25%), no Brassicaceae plants were found.

### 2.2. Emergence Boxes

Brassicaceae collected were put into cardboard emergence boxes of 35 cm × 27 cm × 16 cm. Plants of the same species collected at the same site and date were put together in an emergence box. The number of plants per box was noted and the total number of siliques was counted before introduction into the emergence box. The boxes had an opening leading into a transparent plastic container in which the emerging parasitoids were collected and counted. The boxes were checked 2 to 3 times a week for 60 days to collect parasitoids. After 60 days, the boxes were opened. Weevil larvae were counted, and the number of holes in siliques, stems and root crowns were counted. Parts of the plants with holes were dissected. If a head capsule was present (the remains of parasitized weevil larvae), it was retrieved and preserved in 70% alcohol (Figure 1). The presence of a head capsule also confirmed that the hole had been made by a parasitoid. Weevils and parasitoids were preserved in 70% alcohol. For each box, a quarter of the siliques were dissected, with a minimum of 25 and a maximum of 250 siliques. The presence of weevil larva, meconium, head capsule, or dead parasitoids not emerged was also noted. Larvae, head capsules and non-emerged parasitoids were preserved in 70% alcohol for molecular identification.

### 2.3. Parasitoids and Weevil Identification

The Laboratoire d’expertise et de diagnostic en phytoprotection (LEDP) of the Ministry of Agriculture and Agrifood of Quebec identified the parasitoids and larvae collected. Parasitoids were identified using mostly morphological criteria and identification keys [15]. Damaged specimens of parasitoids were subjected to DNA barcoding. DNA extraction was performed using the DNeasy Blood and Tissue Kit (Qiagen, Hilden, Germany) following the manufacturer’s protocol. Following DNA extraction, amplification of the COI gene was conducted with primers LCO1490 and HCO2198 in reactions containing: 25.00 μL Invitrogen Platinum Hot Start PCR 2× MasterMix, 19.00 μL H_2_O, 0.5 μL 10 μM forward primer, 0.5 μL 10 μM reverse primer, and 5 μL template DNA. Thermal cycler conditions were: 95 °C for 3 min, 45 cycles of 95 °C for 45 s, 53 °C for 45 s, and 72 °C for 45 s, and a final extension of 72 °C for 3 min. Amplification success was verified using QIAxcel Advanced System (Qiagen). Purification and Sanger Sequencing were completed at the SANGER Sequencing Platform of the CHU de Québec-Université Laval Research Center on an ABI 3730xl DNA Analyzer. The sequences were trimmed, aligned, and assembled using Geneious v2021.2.1. We were not able to identify all the specimens of parasitoids even with molecular analysis. Some of them were too damaged. Weevil larvae and head capsules were identified using molecular analysis with the same protocol as described above. Given the large number of larvae and head capsules collected, not all larvae and head capsules samples were sent for molecular analysis.

## 3. Results

A total of 712 plants from 12 Brassicaceae species were collected in 2019 and 2020 and were distributed in 266 emergence boxes. *Ceutorhynchus* were present in 105 boxes (39.5%), but larvae only emerged in 78 boxes (29%) and parasitism was observed in 34 boxes (13%). We were able to identify the *Ceutorhynchus* species in 53 of the 105 boxes that contained weevil larvae. Five species of *Ceutorhynchus* were identified: *C. omissus*, *C. neglectus*, *C. typhae*, *C. erysimi* and *C. obstrictus*. All weevil species were collected from boxes in both years except for *C. erysimi* which was identified only in 2019. Only one *Ceutorhynchus* species was present per box, except for one box in which two species, *C. omissus* and *C. neglectus* were found. We obtained only one identification match for head capsules.

### 3.1. Ceutorhynchus and Their Host Plant

Not all species of Brassicaceae were infested by *Ceutorhynchus*. No *Ceutorhynchus* were found in plants of stinkweed, *Thlaspi arvense* L., yellow rocket, Barbarea vulgaris, common pepper grass, Lepidium densiflorum, and tall wormseed wallflower, Erysimum hieraciifolium. The proportion of Brassicaceae species associated with identified *Ceutorhynchus* varied according to weevil species (Figure 2). All identified *Ceutorhynchus* larvae were directly retrieved from emergence boxes containing only one species of Brassicaceae. With the exception of head capsule samples, all weevil larvae sent for identification came from plants where exit holes of weevil larvae were observed. Most identified specimens of *C. neglectus*, about 65%, were retrieved from marsh yellow cress, Rorippa palustris. This weevil was also retrieved from shepherd’s purse, Capsella bursa-pastoris, wormseed wallflower, and tower rockcress, Arabis glabra. Exit holes of weevil larvae were observed from the three Brassicaceae species mentioned above from which *C. neglectus* larvae were identified. The only head capsule of *C. neglectus* identified was retrieved from wild mustard, Sinapis arvensis, but no exit hole of weevil larvae was observed from this plant. All identified specimens of *C. omissus* were collected from wormseed wallflower, and *Erysimum cheiranthoides*. Most identified specimens of *C. typhae*, about 90%, were retrieved from the shepherd’s purse, and one was retrieved from wormseed wallflower. Only one specimen of *C. erysimi* was identified and was retrieved from the shepherd’s purse. Identified specimens of the canola pest, *C. obstrictus*, were retrieved from three different species of Brassicaceae: bird rape, *Brassica rapa* (syn. *Brassica campestris*), wild radish, *Raphanus raphanistrum*, and wormseed wallflower.

### 3.2. Parasitoids Collected

A total of 146 parasitoids were collected from the boxes and 112 of them (75%) were identified at least at the family level (Figure 3 and Table 1). Parasitoids identified belong to two families of Hymenoptera: Pteromalidae and Eulophidae. Necremnus duplicatus was the only species identified as belonging to the Eulophidae family. All specimens of N. duplicatus emerged from the same box. Chlorocytus, Pteromalus, Trichomalus and Elachertodomyia are genus belonging to the Pteromalidae family. Pteromalus was the most abundant genera representing 50% of all identified parasitoids. The 55 specimens were collected from 24 different samples ranging from 1 to 11 individuals per sample. Chlorocytus is the second most abundant genera representing about 27% of all identified parasitoids, but individuals were only collected from 4 different samples. Number of specimens per sample ranged from 1 to 15.

Only two specimens of Trichomalus perfectus were collected (one in 2019 and one in 2020). Trichomalus perfectus is an important natural enemy of the canola pest *C. obstrictus*. This parasitoid represents 2% of parasitism cases of all identified parasitoids. All specimens of Trichomalus combined represent around 3.5% of parasitism cases if excluding non-identified parasitoids.

### 3.3. Ceutorhynchus Trophic Associations

For most identified parasitoids, we were able to identify their putative host (Figure 4 and Table 1). The native weevil *C. neglectus* was identified as the putative host of all Pteromalidae genera, except the genus Elachertodomyia. It is also the only *Ceutorhynchus* species that was associated with Necremnus duplicatus from the Eulophidae family. All specimens of N. duplicatus emerged from one box containing marsh yellow cress. Four Pteromalus specimens were recovered from *C. neglectus*. Those Pteromalus specimens emerged from marsh yellow cress, tower rockcress, wormseed wallflower and wild mustard and were present in 23% of all samples where *C. neglectus* was present. One Chlorocytus was recovered from *C. neglectus* that emerged from marsh yellow cress. The other 12 specimens of Chlorocytus were recovered from *C. neglectus* that emerged from one sample of tower rockcress. One specimen of T. perfectus and one specimen of Trichomalus lucidus were recovered from *C. neglectus* that emerged from wild mustard and marsh yellow cress respectively. A total of 13 parasitoids from *C. neglectus* reared from three different samples were not identified.

Native *C. omissus* yield 12 specimens of Pteromalidae of which 10 belong to the genus Pteromalus. Four parasitoids recovered from this weevil were not identified. They were collected from 3 different samples. All parasitoids recovered from *C. omissus* that emerged from the exotic weed wormseed wallflower. Pteromalus were yield in 58% of samples where *C. omissus* was identified.

Parasitoids from the Pteromalidae and Eulophidae were reared from weevil species associated with weeds. The exotic weevil *C. erysimi* did not yield any parasitoids, but only one specimen of this weevil species was identified. The exotic weevil *C. typhae* yields a total of 26 specimens of Pteromalidae. They were all collected from the exotic weed shepherd’s purse and a total of 7 Pteromalus were reared from 3 different samples representing 23% of all samples with *C. typhae*. Fifteen specimens of Chlorocytus that were reared from *C. typhae* were collected from the same sample. Three Elachertodomyia from one sample and one T. perfectus from another sample were also reared from *C. typhae*. Two parasitoids associated with *C. typhae* could not be identified.

The canola pest, *C. obstrictus* did not yield any parasitoids and no parasitism was observed in the sample in which this species was identified. The unidentified Trichomalus species was collected from the shepherd’s purse, but no weevil species was identified from this sample. Four specimens of Pteromalus and two specimens of Eulophidae were also collected from the same sample.

## 4. Discussion

*Ceutorhynchus* species identified in Brassicaceae plants are all species that are known to be present in Quebec’s agroecosystem [9]. Except for *C. erysimi* larvae which feed in the root crown, all larvae of the weevil species identified feed in siliques. Larvae of the native *Ceutorhynchus* americanus and exotic *Ceutorhynchus* pallidactylus are present in some regions of Quebec and are known to feed on stems of marsh yellow cress, wild mustard, and/or wild radish [5,9], but they were not identified in this study. Both species are considered rarer species in Quebec’s agroecosystem than the *Ceutorhynchus* species identified in this study [9]. This study confirms previous associations between *Ceutorhynchus*, their host plant, and their parasitoids but also reports new ones. Assemblages of parasitoids observed in this study differed between the years 2019 and 2020. Only specimens of Pteromalus and *T. perfectus* were collected in both years. The overall assemblage observed is dominated by specimens of Pteromalus and Chlorocytus.

From the assemblage of *Ceutorhynchus* species identified, two functional groups can be distinguished: natural enemies of weeds and agricultural pests. The first group is composed of the native *C. omissus* and *C. neglectus*, and the exotic *C. typhae* and *C. erysimi* which are part of the guild of natural enemies of Brassicaceae weeds found in Quebec’s agroecosystem. Although their relative abundance varies among regions, they are common species found in agroecosystems [9].

The native weevil *C. neglectus* was collected from five different host plants and can thus be considered as a narrow generalist [10]. A new association was observed between *C. neglectus* and the native tower rockcress, A. glabra. Exit holes were observed in siliques of tower rockcress, meaning *C. neglectus* larvae were able to complete their development in this host plant. Tower rockcress is not considered as a weed in Quebec’s agroecosystem [16]. This weevil was also associated with exotic weeds shepherd’s purse and wormseed wallflower from which exit holes were also observed. This weevil was also associated with wild mustard, *S. arvensis*, but since the larvae were parasitized and no exit hole of weevil larvae was observed, we cannot assume *C. neglectus* was able to complete its larval development in this plant species. *Ceutorhynchus* neglectus was the only weevil species identified that has both native and exotic host plants. The highest densities of this weevil were observed in marsh yellow cress and tower rockcress, its native host plants.

*Ceutorhynchus* neglectus was associated with the greatest number of parasitoids taxon (5) from the Pteromalidae and the Eulophidae family. Most parasitoid specimens associated with *C. neglectus* were Chlorocytus, but Pteromalus were found in the greatest number of samples. This weevil was associated with the Holarctic *T. lucidus*, the exotic *T. perfectus*, and the native species *N. duplicatus*.

The native *C. omissus* was associated with one host plant, the exotic weed wormseed wallflower, *E. cheiranthoides*. Previous studies and our results showed that *C. omissus* is a specialist in this host plant in Quebec’s agroecosystem [5,9]. This weevil hosted only parasitoids from the Pteromalidae family. This weevil also had the greatest proportion of parasitism cases associated with Pteromalus.

The exotic *C. erysimi* was retrieved from the root crown of a Shepherd’s purse, C. bursa pastoris, its known host plant [5]. Only one specimen was identified, and no parasitism was observed in the associated sample.

The exotic weevil *C. typhae* was associated with two host plants: exotic weed wormseed wallflower, and Shepherd’s purse. Association between wormseed wallflower and *C. typhae* is reported for the first time. Since only one identified specimen of *C. typhae* was collected from this host plant, this insect-host plant association may be incidental. The host plant associations of *C. erysimi* and *C. typhae* were restricted to one or two hosts with a more significant association for one of them, making those weevil species more specialist than a generalist. Being specialists and relatively abundant in Quebec’s agroecosystem [5,17], these two species are interesting candidates for the biological control of the exotic weed Shepherd’s purse. It would be however important to consider the impact of parasitoids on those species if their potential as biocontrol agents is further investigated.

*Ceutorhynchus* typhae hosted parasitoids from the Pteromalidae and Eulophidae families. Association between *C. typhae* and the Pteromalidae Elachertodomyia, was observed for the first time in Eastern Canada, and, to our knowledge, it has never been reported elsewhere in the literature. Although the greatest number of parasitoids associated with this weevil was *Chlorocytus*, *Pteromalus* were found in more samples.

The second functional group is only composed of the cabbage seedpod weevil, *C. obstrictus*, an important pest of canola [18,19]. Larvae of *C. obstrictus* completed their development in two exotic weeds: bird rape, *B. rapa* (also known as turnip rape), and wormseed wallflower. Larvae of *C. obstrictus* were also retrieved from the silique of wild radish, *R. raphanistrum*, but no exit hole was observed from these plants. Association between *C. obstrictus* and wild radish has been reported in Quebec by Mason et al. [20]. An association between bird rape and *C. obstrictus* has been observed [20] but has never reported in Quebec before. Although bird rape is the same species as polish type canola (*B. rapa*), it has not been cultivated and is considered a weed in Quebec [21]. Polish-type canola is not cultivated in Quebec. Bird rape was an effective trap crop in Alberta [22] but did not prove to be effective to manage the cabbage seedpod weevil and other *Ceutorhynchus* pests found in rapeseed [23,24]. Wormseed wallflower has never been reported before as a host plant of *C. obstrictus*. Since only one specimen was reared from wormseed wallflower, the association between *C. obstrictus* and this Brassicaceae may be incidental. This weevil is known to feed on volunteer canola and other wild Brassicaceae in spring, but as our results and other studies show, it can also complete its development in other wild Brassicaceae species [19,22,25]. The presence of host plants other than canola suggests that crop rotation may not be sufficient to manage populations of *C. obstrictus* and weed management should be considered [26]. However, since wild Brassicaceae may constitute an important resource for insect pollinators and further assessment of *C. obstrictus* abundance, its impact should be assessed before implementing control of these wild Brassicaceae.

No parasitoids were collected from samples with *C. obstrictus*. In western Canada, *C. obstrictus* was parasitized by Trichomalus lucidus, Chlorocytus sp and Pteromalus sp. in canola fields which are species and genus also present in Quebec [27]. Although no parasitism on *C. obstrictus* was observed in areas bordering canola and other crops, it is possible that this weevil is parasitized by some of these parasitoids in canola fields. Two specimens of T. perfectus, an important natural enemy of *C. obstrictus* [28,29], were identified, but were collected from other *Ceutorhynchus* species. This parasitoid was abundant in many canola fields of Quebec in 2019 and 2020 adjacent to sampling sites where wild Brassicaceae were collected [30]. Since T. perfectus is not abundant in areas adjacent to or near canola fields, those areas where alternative host plants are present may serve as a refuge for *C. obstrictus* [31]. The cabbage seedpod weevil is thus free from its natural enemies in those areas while other *Ceutorhynchus* species are sometimes highly parasitized [32]. This situation could bring a selective advantage to *C. obstrictus*. Since T. perfectus has been reported in 2009 as emerging from canola siliques of Ontario and Quebec, concerns about its impact on non-target weevil species have been raised [14,33,34]. Our results show that T. perfectus is quite rare in the parasitoid assemblage associated with *Ceutorhynchus*, but it can attack native weevil such as *C. neglectus* and other natural enemies of weeds such as *C. typhae*. The exotic parasitoid Mesopolobus gemellus was a dominant species in the parasitoid assemblage observed by Mason et al. [5] from 2006 to 2011 in Quebec and Ontario, but no specimens of this species have been identified in the present study.

Previous studies have found a greater diversity of parasitoids associated with *Ceutorhynchus*, but those studies used different methods and areas sampled, and the time frame of the sampling differed [5,14]. It is therefore difficult to compare previous studies with this one. To have a complete, optimal, and objective portrait, the study should be extended over a higher number of years and the sampling effort should be intensified.

Concluding, the characterization of trophic associations of *Ceutorhynchus*, their host plants, and their parasitoids allowed us to identify new associations and confirmed some that had been already observed. Determination of parasitoid assemblages enabled us to observe the spread and abundance of exotic parasitoids. Furthermore, this characterization provided useful insight and tools in identifying new potential biological control agents against Brassicaceae weeds and managing the canola pest *C. obstrictus* in relation to its association with wild Brassicaceae. It will also provide evidence in the future to study the role of some of these host plants in the invasion process of these introduced parasitoids. The issue of pest weevils in Brassicaceae crops (camelina, mustard, and canola) cannot be resolved without an extensive understanding of the distribution and the abundance of wild Brassicaceae, their associated *Ceutorhynchus* weevils, and their parasitoids.

## Figures and Tables

**Figure 1 insects-14-00607-f001:**
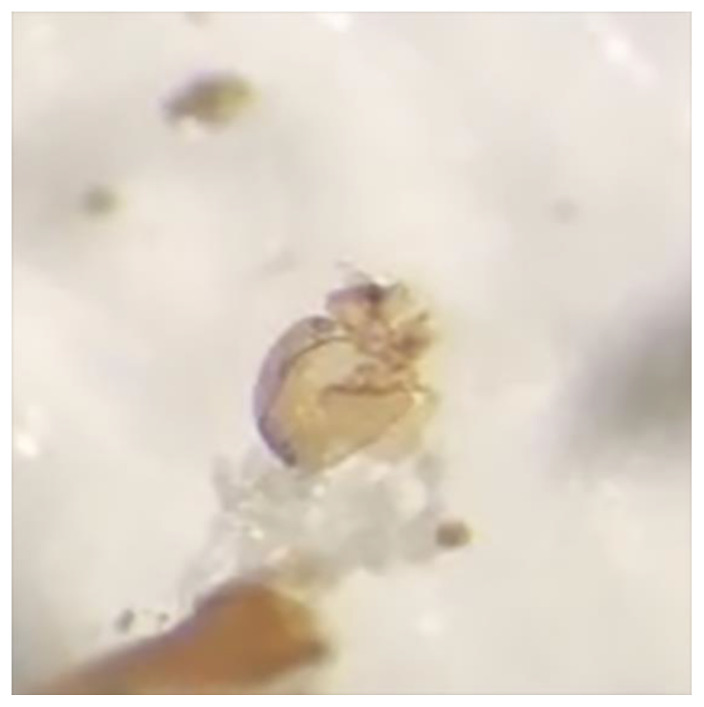
Head capsule of parasitized *Ceutorhynchus* larvae collected in a silique of wormseed wallflower, *Erysimum cheiranthoides*.

**Figure 2 insects-14-00607-f002:**
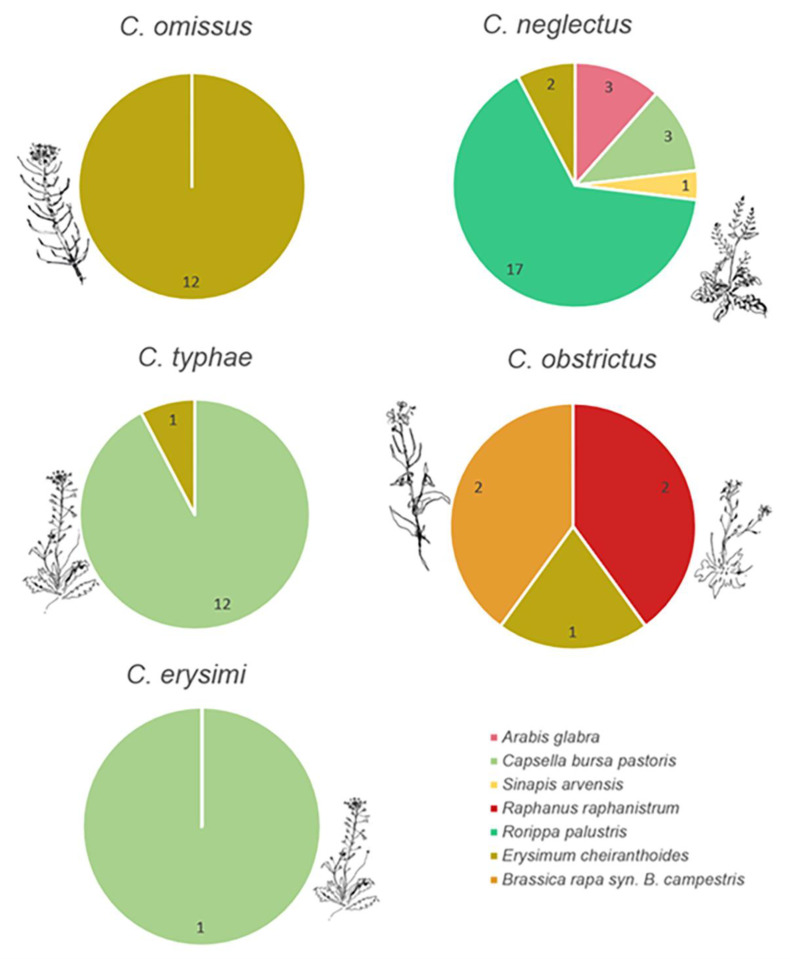
Occurrence of Ceuthorynchus larvae per *Brassicaceae* species sampled in 2019 and 2020. Number indicates counts of boxes with the *Brassicaceae* species and the *Ceutorhynchus* species.

**Figure 3 insects-14-00607-f003:**
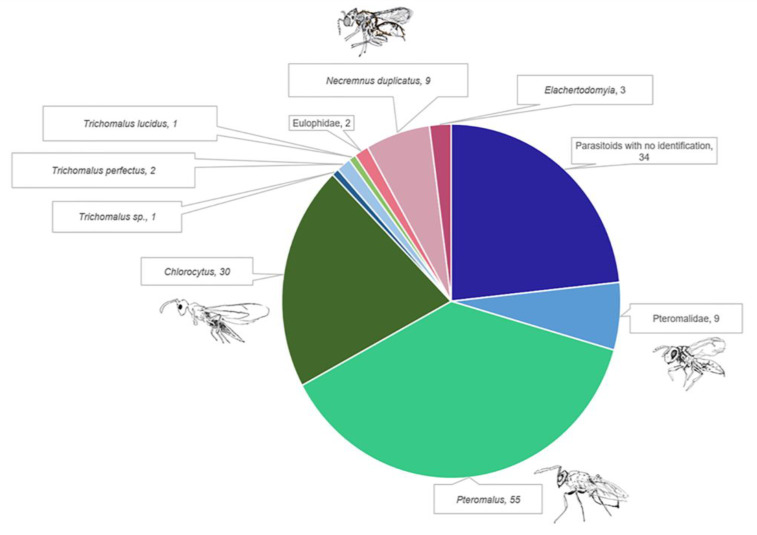
Composition of the parasitoid assemblage that emerged from Brassicaceae plants collected in 2019 and 2020 in six regions of Quebec (Bas-Saint-Laurent, Capitale-Nationale, Centre-du-Québec, Chaudière-Appalache, Montérégie and Saguenay Lac-Saint-Jean).

**Figure 4 insects-14-00607-f004:**
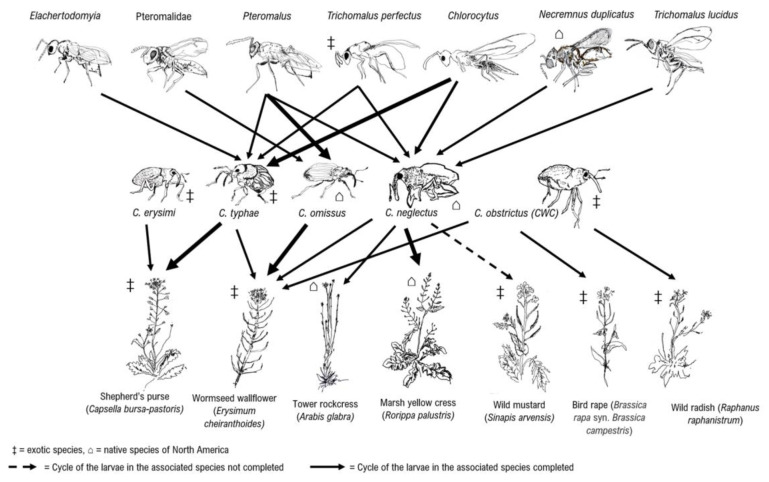
Host plant associations of *Ceutorhynchus* and their parasitoids. ‡ = exotic species, ⌂ = native species of North America. The thicker lines for relationships that have been seen more times.

**Table 1 insects-14-00607-t001:** Parasitoids reared from *Ceutorhynchus* hosts collected from Brassicaceae plant hosts in 2019 and 2020.

	*C. neglectus*				*C. omissus*	*C. typhae*	Unidentified *Ceutorhynchus*		
	*Rorippa* *palustris* Marsh Yellow Cress	*Sinapis* *arvensis* Wild Mustard	*Erysimum* *cheiranthoides* Wormseed Mustard	*Arabis glabra* Tower Rockcress	*Erysimum* *cheiranthoides* Wormseed Mustard	*Capsella* *bursa-pastoris* Shepherd’s Purse	*Rorippa* *palustris* Marsh Yellow Cress	*Capsella* *bursa-pastoris* Shepherd’s Purse	*Erysimum* *cheiranthoides* Wormseed Mustard
**Pteromalidae**									
*Chlorocytus* species	1			12		15			2
*Pteromalus* species	1	1	1	1	11	7	1	4	22
*Trichomalus lucidus* (Walker)	1								
*Trichomalus perfectus* (Walker)		1				1			
*Trichomalus* species								1	
Unidentified Pteromalidae species					2				7
*Elachertodomyia* species						3			
**Eulophidae**									
*Necremnus duplicatus* Gahan	9								
Unidentified Eulophidae species								2	

## Data Availability

The data presented in this study are available in the article.

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
