# Peer review of "Trophic Interactions of Ceutorhynchinae spp. (Coleoptera: Curculionidae) with Their Host Plants (Brassicaceae) and Their Parasitoids in the Agroecosystem of Quebec, Canada"

_insects, 2023, doi:10.3390/insects14070607_

Round 1
Author Response
Review of Insects-2386666
The authors examined the densities of weevils and their parasitoids on host plants in the Brassicaceae in Quebec. They describe the trophic relationships among host plants, weevils, and the parasitoids of the weevils. The study motivation and design is sound, and the conclusion reached are reasonable and interesting. However, there are a number of items the author's could improve to clarify their presentation and solidify their results.
Data analysis
My biggest concern is with the data analysis. Unless the number of siliques, stems, and root crowns were standardized in each box, then some of the differences in weevil abundance and subsequently parasitoid abundance among replicate boxes will arise because of the different amounts of plant material. Furthermore, the same number of each plant module (e.g., siliques, stems, and root crowns) mist be inspected for weevils in each box. The Figures suggest the authors based their analysis on the total number of individuals of each species in each box. This would be find if the amount of plant material in each box is standardized. If not, the weevil abundance should be estimated a s number per silique or number per root crown, etc. These would be estimates of abundance corrected for the availability of plant material , or an estimate of density. The authors should either clarify what they did, and if plant material was not standardized, then replace the analysis and Figures in the current draft, with revised results based on true densities (number/silique, etc).
Response: the section was removed. No mention of weevil density. No statistics are included in the article. Only descriptive results.
The analysis of percent parasitism is fine since the number of parasitoids is normalized for the number of hosts.
The authors also do not describe the methods they applied to conduct their post hoc multiple comparisons. This should be included in the methods. The authors should also discuss how they control the false positive rate which increases when conducting these sorts of multiple comparisons procedures.
Response: the section was removed. No mention of weevil density. No statistics are included in the article. Only descriptive results.
The authors often get singular and plural forms of words confused. For example, if discussing more than one use "genera," if discussing a single use "genus." On line 63, it should be "parasitoid community," not "parasitoids." In this instance, community is singular even if it is comprised by several species of parasitoids. This problem occurs throughout the manuscript.
Response: I revised it at my best.
Other small issues
line 259 "yield" not yielded
Response: Corrected
line 343 "no exit hole" not "not exit hole"
Response: Corrected
line 81 "Sinceno" not familiar with this term please clarify in text
Response: Corrected
Figure 1, explain the number in the Figure in the figure caption.
Response: No number in the caption of Figure 1, but see caption of Figure 2
Figure 3 and Figure 5, explain the different color bars. I assume there indicate 2019 and 2020 data, but there is no explanation in the Figure or in the caption.
Response: Figures deleted
Figure 6 - More common to represent a food web with the arrow point from higher to lower trophic levels. The authors should consider revising Figure 6, just by placing the arrow head on the opposite ends of each line pointing from parasitoid to weevil host and from weevil to host plant. The line then indicating which weevils are attached by which parasitoid and which host plants are fed upon by which weevils.
Response: See the figure in question. I changed the direction of the arrow head. However, trophic figures are also often represented with arrow pointing in the direction of the energy flow.
Reviewer 2 Report
Trophic interactions of Ceutorhynchinae Spp. (Coleoptera: Cucurlionidae) with their host plants (Brassicaceae) and their par asitoids in the agroecosystem of Quebec, Canada
The authors studied the trophic interactions between five different Ceutorhynchus species, its host plants and its parasitoids, and assessed the ecological role of these insects in the agricultural landscape of Eastern Canada. They also looked at the level of plant infestation by Ceutorhynchus species, and their parasitism levels. They showed novel associations between Ceutorhynchus typhae and the parasitoids Elachertodomyia, or between Ceutorhynchus obstrictus and the host plants bird rape and wormseed wallflower; and confirmed associations that had been already observed.
The authors did a great sampling effort (6 regions, 8 sites per region, 5 plants per site, from beginning of July to the end of August, 2 to 5 times per year) with a total of 712 plants over the two years.
This is a timely and important topic as there are few studies on the trophic relations and ecology of Ceutorhynchus, which includes economically important canola pests, and its parasitoids in Canada. Therefore, I think this manuscript is suitable for publication in Insect journal with minor corrections (detailed below detailed comments).
Detailed comments
L44: you can add something like, when ready to pupate they open a hole in the silique/pod, stems, or root crown, the larvae exit the plant and pupate in the soil. This way it links with why you count holes to assess their density.
L63-68: In this paragraph I would recommend add a bit of the parasitoid's life cycle and the holes they do in the plant when exiting.
L80: could you explain this a bit more? why the regions were visited a different number of times? how did you choose which ones were samples 5 times and which ones less times?
L81: “since no”, needs a space between words
L106: could you explain the difference between weevils’ holes and parasitoids holes?
L109: why did you choose to estimate parasitism by dividing the parasitoids holes by the total number of holes? Do parasitoids only do one hole per host – could you add references to this? Wouldn’t it make more sense to do parasitoid density/weevil density to calculate the parasitism rate?
L142: why could you not identify the other Ceutorhynchus spps?
L158: can you add the scientific name of wormseed wallflower?
L173: merge these sentences, it is really the last one that gives the important information
L177: because of the way it was calculated the graph doesn't make much sense to me. I expect that plants with similar density of weevils and parasitoids have higher parasitism, however, is not
L182: what is the sample size? can you add it in brackets next to "small samples sizes"
L183: (Figure3) why per box and not per plant as it was recorded? there were not the same number of plants per box right? therefore this should be per plant to make sense
L193 and 195: there are double spaces before the scientific names
L198: when you say per samples you mean per box? per site? per plant?
L215-219: shouldn't this be in section 3.2 weevil and parasitoids densities?
L229: (Figure 5) again why per rearing box? why not per plant?
L243: I think in this sentence you wanted to say C. typhae instead of C. neglectus right? according to the figure below (figure 6).
L251: (Figure 6) this figure is great to understand the interactions between parasitoids, insect pest and host plant. But I think it will improve if the connection lines reflect the "strength" of the relationship - thicker lines for relationships that have been seen more times, for example Chlorocystus with C. neglectus was only detected ones, but it was detected 12 times with C. typhae.
L314: Quebec's agroecosystem
L340: pest of canola
L348: repeated sentence
Author Response
Trophic interactions of Ceutorhynchinae Spp. (Coleoptera: Cucurlionidae) with their host plants (Brassicaceae) and their par asitoids in the agroecosystem of Quebec, Canada
The authors studied the trophic interactions between five different Ceutorhynchus species, its host plants and its parasitoids, and assessed the ecological role of these insects in the agricultural landscape of Eastern Canada. They also looked at the level of plant infestation by Ceutorhynchus species, and their parasitism levels. They showed novel associations between Ceutorhynchus typhae and the parasitoids Elachertodomyia, or between Ceutorhynchus obstrictus and the host plants bird rape and wormseed wallflower; and confirmed associations that had been already observed.
The authors did a great sampling effort (6 regions, 8 sites per region, 5 plants per site, from beginning of July to the end of August, 2 to 5 times per year) with a total of 712 plants over the two years.
This is a timely and important topic as there are few studies on the trophic relations and ecology of Ceutorhynchus, which includes economically important canola pests, and its parasitoids in Canada. Therefore, I think this manuscript is suitable for publication in Insect journal with minor corrections (detailed below detailed comments).
Detailed comments
L44: you can add something like, when ready to pupate they open a hole in the silique/pod, stems, or root crown, the larvae exit the plant and pupate in the soil. This way it links with why you count holes to assess their density.
Response: See the added sentence in the paragraph.
L63-68: In this paragraph I would recommend add a bit of the parasitoid's life cycle and the holes they do in the plant when exiting.
Response: See the added sentences in the paragraph.
L80: could you explain this a bit more? why the regions were visited a different number of times? how did you choose which ones were samples 5 times and which ones less times?
Response: It is simply because we based in Montreal and we were not able to visit the three other regions more than twice per season. See the added sentence. I added a sentence.
L81: “since no”, needs a space between words
Response: Corrected
L106: could you explain the difference between weevils’ holes and parasitoids holes?
Response: See the added sentences.
L109: why did you choose to estimate parasitism by dividing the parasitoids holes by the total number of holes? Do parasitoids only do one hole per host – could you add references to this? Wouldn’t it make more sense to do parasitoid density/weevil density to calculate the parasitism rate?
Response: Section deleted
L142: why could you not identify the other Ceutorhynchus spps?
Response: We had budget restrictions of the number of samples we could identified and some molecular identifications were not precise enough to identify the sample to the species level.
L158: can you add the scientific name of wormseed wallflower?
Response: It is already present in the article, at the first mention.
L173: merge these sentences, it is really the last one that gives the important information
Lines numbers do not match the ones I have on my version. I don’t know which sentence the reviewer is referring to here.
L177: because of the way it was calculated the graph doesn't make much sense to me. I expect that plants with similar density of weevils and parasitoids have higher parasitism, however, is not
Response: Figure deleted
L182: what is the sample size? can you add it in brackets next to "small samples sizes"
Response: Section deleted
L183: (Figure3) why per box and not per plant as it was recorded? there were not the same number of plants per box right? therefore this should be per plant to make sense
Response: Section deleted
L193 and 195: there are double spaces before the scientific names
Response: Corrected
L198: when you say per samples you mean per box? per site? per plant?
Response: Section deleted
L215-219: shouldn't this be in section 3.2 weevil and parasitoids densities?
Response: Section deleted
L229: (Figure 5) again why per rearing box? why not per plant?
Response: Section deleted
L243: I think in this sentence you wanted to say C. typhae instead of C. neglectus right? according to the figure below (figure 6).
Response: Specimens instead of species
L251: (Figure 6) this figure is great to understand the interactions between parasitoids, insect pest and host plant. But I think it will improve if the connection lines reflect the "strength" of the relationship - thicker lines for relationships that have been seen more times, for example Chlorocystus with C. neglectus was only detected ones, but it was detected 12 times with C. typhae.
Response: See the figure 4
L314: Quebec's agroecosystem
Response: Corrected
L340: pest of canola
Response: Corrected
L348: repeated sentence
Response: Lines numbers do match what’s in the text
I have decided to take out the section about weevil density, parasitoids density and percentage of parasitism. Although I think my data are valid and that field data is rare, I feel it would need more work and time to correctly explain the method used and how to interpret it. I unfortunately do not have the time to work more on these data.